# Online Measurement of Melt-Pool Width in Direct Laser Deposition Process Based on Binocular Vision and Perspective Transformation

**DOI:** 10.3390/ma17133337

**Published:** 2024-07-05

**Authors:** Yanshun Lu, Muzheng Xiao, Xiyi Chen, Yuxin Sang, Zongxin Liu, Xin Jin, Zhijing Zhang

**Affiliations:** 1Department of Mechanical Engineering, Beijing Institute of Technology, Beijing 100081, China; 3120220370@bit.edu.cn (Y.L.); sangyx0825@163.com (Y.S.); 18842378107@163.com (Z.L.); goldking@bit.edu.cn (X.J.); zhzhj@bit.edu.cn (Z.Z.); 2Department of Information and Electronics, Wuhan Digital Engineering Institute, Wuhan 430074, China; seexy0823@gmail.com

**Keywords:** direct laser deposition, melt-pool width, machine vision, image processing, online measurement

## Abstract

Direct laser deposition (DLD) requires high-energy input and causes poor stability and portability. To improve the deposited layer quality, conducting online measurements and feedback control of the dimensions, temperature, and other melt-pool parameters during deposition is essential. Currently, melt-pool dimension measurement is mainly based on machine vision methods, which can mostly detect only the deposition direction of a single melt pool, limiting their measurement range and applicability. We propose a binocular-vision-based online measurement method to detect the melt-pool width during DLD. The method uses a perspective transformation algorithm to align multicamera measurements into a single-coordinate system and a fuzzy entropy threshold segmentation algorithm to extract the melt-pool true contour. This effectively captures melt-pool width information in various deposition directions. A DLD measurement system was constructed, establishing an online model that maps the melt-pool width to the offline deposited layer width, validating the accuracy of the binocular vision system in measuring melt-pool width at different deposition angles. The method achieved high accuracy for melt-pool measurements within certain deposition angle ranges. Within the 30°–60° measurement range, the average error is 0.056 mm, with <3% error. The proposed method enhances the detectable range of melt-pool widths, improving cladding layers and parts.

## 1. Introduction

Direct laser deposition (DLD) is an advanced manufacturing technology that utilizes high-energy lasers to melt and deposit metal powders simultaneously onto three-dimensional parts [1]. This technique offers advantages such as superior part performance, high manufacturing flexibility, short production cycles, and low costs, attracting significant attention from both academia and industry [2]. However, as a complex process with numerous variables [3], DLD faces challenges, such as poor process continuity and limited portability. Therefore, online measurements of the melt pool during deposition are essential for improving quality and yield.

The dimensions of the melt pool significantly affect the morphology of the deposited layer. During DLD, the melt pool exhibits high brightness, high temperature, small size, and rapid changes, with interference from sparks and plasma. Noncontact sensors based on machine vision offer a mainstream solution to these measurement challenges, allowing precise acquisition of essential melt-pool data.

Thermal and infrared imaging are the most common noncontact measurement methods. Criales et al. [4] used a thermal camera to observe the geometrical features and splattering behavior of nickel alloy 625 during laser-powder bed fusion (L-PBF). Hu and Kovacevic employed [5] a coaxial camera to capture infrared images of the melt pool, with the infrared signals fed back to a control system to regulate laser power and maintain a stable melt-pool width. Price et al. [6] captured melt-pool images in electron beam additive manufacturing (EBAM) of Ti-6Al-4V powder using near-infrared (NIR) thermal imaging to analyze the temperature distribution and dimensions of the melt pool. Heigel and Lane [7] identified liquid–solid transition discontinuities in an L-PBF melt pool by utilizing the local minima of the second derivative of the grayscale intensity signals captured by an IR high-speed camera. They then determined the overall shape of the melt pool by analyzing the corresponding intensity values. Cheng et al. [8] collected radiation temperature data with an NIR thermal imager during selective laser melting and utilized the collected thermal images and radiation temperature curves to calculate the liquid–solid phase transition lines and extract melt-pool width information. Da Silva et al. [9] used a laser coaxial thermal imager to detect overhead melt-pool contours and extract geometric dimensions, such as length and width. However, the high cost of high-speed, high-resolution NIR cameras limits their practical application.

With advancements in semiconductor devices and integrated circuit technology, relatively low-cost image sensors, such as charge-coupled devices (CCDs) and complementary metal-oxide semiconductors (CMOSs) have become widely used. Machine-vision-based melt-pool image measurements primarily focus on dimensions, such as the width and height of the melt pool. Kim and Ahn [10] developed a monitoring system with a coaxial illumination laser and a CCD camera equipped with optical filters designed to monitor the two-dimensional keyhole shape in Yb: YAG laser welding. Doubenskaia et al. [11] demonstrated that in the Ti6Al4V laser cladding process, grayscale image brightness in the liquid–solid transition zones is consistent across different cladding parameters, providing a reliable method for estimating temperature distribution and calculating dimensions of the melt pool. Yang et al. [12] developed an online system using a side-positioned CCD camera with neutral-density filters and image-processing algorithms to eliminate noise and accurately determine melt-pool width. Hao et al. [13] introduced a weld pool imaging system based on spatial filtering and the Abbe imaging theory, using a high-pass spatial filter to eliminate low-frequency background noise and precisely capture the contours of the melt pool, weld seam, and arc. Le et al. [14] established a machine-vision-based laser selective melting system for online melt-pool dimension measurement with a lateral off-axis setup. Devesse et al. [15] developed a field programmable gate array-based image capture system that transferred sensor data to a computer for processing with MATLAB simulations to predict melt-pool dimensions. However, these single-camera side-shot methods are limited to detecting melt pools in one deposition direction.

Asselin et al. [16] used a trinocular vision system with cameras set 120° apart to monitor the melt pool from various angles, applying perspective transformation algorithms to measure melt-pool widths from different directions; however, they overlooked edge position changes due to varying deposition directions. Mazzoleni et al. [17] proposed a coaxial imaging system with an external light source monitor module, CMOS camera, and filters. Yang [18] designed a coaxial camera system that measured melt-pool dimensions by capturing the entire upper surface of the melt pool with a coaxially aligned camera and using the minimum bounding rectangle method for dimension extraction.

Recently, machine-learning-based neural network techniques have been applied to the measurement of melt-pool images. Yu et al. [19] utilized an enhanced U-Net architecture network trained with limited sample data for real-time recognition and tracking of arc-welding melt-pool boundaries. Wang et al. [20] input melt-pool images and simultaneously collected weld voltage signals transformed via short-time Fourier transform (STFT) into a CNN to identify welding defects. However, these neural networks only achieved target detection and were significantly affected by noise, reducing accuracy in melt-pool dimension measurements. Jiang et al. [21] developed a backpropagation neural network prediction model for laser cladding height using process parameters (laser power, powder feed rate, and scanning speed) to predict melt-pool height. Building on the work by Asselin et al. [16], Ravani-Tabrizipour et al. [22] input information, such as the angles of the major and minor axes of the melt-pool ellipse, obtained from multicamera projection transformations, into an algorithm combining image tracking protocols and recursive neural networks to detect the overlay height of SS303L deposited on low-carbon steel under various conditions. However, the output represents the height of the deposited overlay, which differs from that of the liquid melt pool during deposition. She et al. [23] built a multisensor monitoring system for the laser welding process, extracted the image features of the molten pool as well as laser-induced plasma spectral features, and established a real-time prediction model for the high-precision welding depth of fusion.

Most machine-vision-based methods for detecting melt-pool width [12,13,14,15] rely on off-axis side-mounted monocular cameras, which are practical only for melt pools deposited along the optical axis of the camera, limiting their utility. When the deposition direction of the melt pool is not parallel to the camera setup, the limitations of the camera’s projection detection principle result in detecting only the near edge while the far edge is obscured, capturing a “false edge” (Figure 1) projection on the upper surface of the melt pool.

To improve the measurement range and accuracy of melt-pool width and prevent incorrect width extraction due to false edges, this study proposes a binocular-vision-based method employing off-axis cameras. Binocular cameras detect both edges of the melt pool, and the images are aligned in the same spatial plane coordinate system using perspective and affine transformations. By incorporating deposition angle data, this method achieves accurate melt-pool width measurements.

## 2. Proposed Binocular Vision Measurement Method

### 2.1. Principle of Melt-Pool Width Measurement

Figure 2 shows the principle of binocular vision melt-pool width measurement. Two vertically aligned, side-mounted cameras capture the true edges of the melt pool.

As shown in Figure 2a, the cameras are positioned around the direction of melt-pool deposition: Camera 1 detects the true lower edge, while Camera 2 captures the true upper edge. After applying perspective transformation and image processing, the images from both cameras are aligned in the same spatial plane. While the loss of some edges at the very front of the melt pool can result in an incomplete reconstruction of the entire melt-pool contour, the edges at the widest part of the melt pool remain effectively preserved. Therefore, by using the true edge information from each camera, the true width of the melt pool can be extracted, as illustrated in Figure 2b.

### 2.2. Perspective Transformation and Spatial Resolution Calibration

To align the images captured by the binocular cameras into the same spatial plane, the images were transformed from the camera coordinate system to the coordinate system of the DLD melt pool using perspective transformation principles. As shown in Figure 3, this transformation converts the side-view images of the melt pool into top-view images, simplifying the extraction of the melt-pool width. Prior to the experiment, pixel response calibration of the CMOS image sensors was performed using a flat-field model on a speckled region scale [24].

As described in [25], the perspective transformation process is conducted in accordance with
(1)$$\left[\begin{array}{ccc} x^{\prime } & y^{\prime } & w^{\prime } \end{array}\right]=\left[\begin{array}{ccc} u & v & w \end{array}\right]\left[\begin{array}{ccc} a_{11} & a_{12} & a_{13}\\ a_{21} & a_{22} & a_{23}\\ a_{31} & a_{32} & a_{33} \end{array}\right],$$
where (u,v) is the coordinates of the original image corresponding to the transformed image coordinate (x,y), and w is the third coordinate. Without loss of generality, the values of w and $a_{33}$ are set to 1. The expressions of x and y are expressed as follows:(2)$$x=\frac{x^{\prime }}{w^{\prime }},\;y=\frac{y^{\prime }}{w^{\prime }}$$

Transforming the system of equations yields the following expression:(3)$$x=\frac{x^{\prime }}{w^{\prime }}=\frac{a_{11}u+a_{21}v+a_{31}}{a_{13}u+a_{23}v+a_{33}}\;,$$
(4)$$y=\frac{y^{\prime }}{w^{\prime }}=\frac{a_{12}u+a_{22}v+a_{32}}{a_{13}u+a_{23}v+a_{33}}\;.$$

The perspective transformation matrix involves eight unknown parameters and requires eight equations, implying that the coordinates of four points are required to solve the problem. The system processes images to determine these coordinates by placing the calibration board in a predetermined orientation within both the cameras’ views and captured images. Solving this system of equations allows for the calculation of the transformation matrix. Additionally, pixel equivalence is determined using the distance between the corners and the number of image pixels, as shown in the following formula, where Lreal represents the real size of calibration board, and $L_{pixel}$ represents the pixel size in the corresponding image.
(5)$$Resolution\left[\frac{\mathrm{mm}}{\mathrm{pixel}}\right]=\frac{Lreal_{\left[\mathrm{mm}\right]}}{L_{pixel}\left[\mathrm{pixel}\right]}.$$

### 2.3. Image Processing Algorithms

#### 2.3.1. Image Denoising

In the melt-pool target detection system described in [19], noise removal was unnecessary because the system did not calculate melt-pool dimensions. However, the DLD process in this study encounters noise from sparks and plasma, significantly affecting melt-pool measurement accuracy. To address this, we utilized a bilateral filter to mitigate spark noise and applied a multilevel thresholding algorithm based on fuzzy entropy to counteract plasma effects, thereby accurately delineating the true edges of the melt pool.

The bilateral filter, a nonlinear filter, extends the Gaussian spatial distance filter by incorporating image grayscale values. The weight function is expressed as follows:(6)$$\begin{array}{cc} W_{s}= & \; \end{array}e^{-\frac{{\left(x_{j}-x_{i}\right)}^{2}+{\left(y_{j}-y_{i}\right)}^{2}}{2{\sigma_{s}}^{2}}},$$
(7)$$\begin{array}{cc} W_{r}= & \; \end{array}e^{-\frac{{\left(gray\left(x_{j},y_{j}\right)-gray\left(x_{i},y_{i}\right)\right)}^{2}}{2{\sigma_{r}}^{2}}}$$
(8)$$\begin{array}{cc} W_{ij}= & \frac{1}{K_{i}} \end{array}*W_{s}*W_{r},$$
where $gray\left(x_{j},y_{j}\right)$ represents the grayscale value at the current point and $gray\left(x_{i},y_{i}\right)$ represents the grayscale value at the center point. $W_{s}$ is the spatial domain weight, $W_{r}$ is the range domain weight, $\sigma_{r}$ is the standard deviation of the spatial domain, and $\sigma_{s}$ is the standard deviation for grayscale values. In bilateral filtering, the spatial domain function removes the noise contamination, whereas the range domain function preserves the high-frequency details of the image. Figure 4 shows a comparison of the melt-pool images obtained using various filters.

#### 2.3.2. Melt-Pool Contour Segmentation

Traditional single-threshold segmentation methods are insufficient for achieving complete separation to effectively mitigate the impact of plasma noise and accurately segment the melt pool from the image, especially considering the continuity between the melt pool and plasma images. Therefore, this study introduces a multilevel threshold segmentation method based on fuzzy entropy.

The total entropy of the image [26] is defined as follows, where P represents the normalized histogram distribution of the image with L=255 gray levels, and $p_{i}$ represents the probability of the i-th gray level occurring in the entire image.
(9)$$\;H\left(P\right)=-\overset{n}{\underset{i}{\sum }}p_{i}lnp_{i}$$

Suppose the image has *n* − 1 thresholds that divide the normalized histogram into n classes. Employing a trapezoidal membership function, the maximum fuzzy entropy for each segment across *n* levels is defined as follows:(10)$$H_{1}=-\overset{L-1}{\underset{i=0}{\sum }}\frac{p_{i}*\mu_{1}\left(i\right)}{P_{1}}*\ln\left(\frac{p_{i}*\mu_{1}\left(i\right)}{P_{1}}\right)$$
(11)$$H_{2}=-\overset{L-1}{\underset{i=0}{\sum }}\frac{p_{i}*\mu_{2}\left(i\right)}{P_{2}}*\ln\left(\frac{p_{i}*\mu_{2}\left(i\right)}{P_{2}}\right)$$
(12)$$H_{n}=-\overset{L-1}{\underset{i=0}{\sum }}\frac{p_{i}*\mu_{n}\left(i\right)}{P_{n}}*\ln\left(\frac{p_{i}*\mu_{n}\left(i\right)}{P_{n}}\right)$$
where
(13)$$P_{1}=\overset{L-1}{\underset{i=0}{\sum }}p_{i}*\mu_{1}\left(i\right),P_{2}=\overset{L-1}{\underset{i=0}{\sum }}p_{i}*\mu_{2}\left(i\right),\ldots ,P_{n}=\overset{L-1}{\underset{i=0}{\sum }}p_{i}*\mu_{n}\left(i\right)$$

Optimal parameters are determined by identifying the maximum total entropy:(14)$$\varphi \left(a_{1},c_{1}\ldots a_{n-1},c_{n-1}\right)=Argmax\left(\left[H_{1}+H_{2}+\ldots +H_{n}\right]\right)$$

The parameter optimization is performed using the differential evolution method with the following settings: population size sizepop=60; variable dimension vardim=6 for three-level thresholding; variable boundaries bound=[0, 255]; crossover and mutation rates of [0.8, 0.6]. By varying $a_{1},c_{1}\ldots a_{n-1},c_{n-1}$ to maximize the function $\varphi \left(a_{1},c_{1}\ldots a_{n-1},c_{n-1}\right)$, the optimized parameters can be obtained, from which the n−1 thresholds can be derived using the following equation:(15)$$t_{1}=\frac{\left(a_{1}+c_{1}\right)}{2},t_{2}=\frac{\left(a_{2}+c_{2}\right)}{2},\ldots ,t_{n-1}=\frac{\left(a_{n-1}+c_{n-1}\right)}{2}$$

Figure 5 compares the effectiveness of the multilevel thresholding method proposed in this study with the Otsu single-level thresholding algorithm, which seeks to maximize interclass variance based on grayscale traversal. The three-level thresholding successfully separates reflections, plasma, and the actual melt-pool contours, demonstrating a superior segmentation capability.

### 2.4. Dimension Extraction of Melt Pools

After segmentation, the melt-pool contour shifted in a direction within the perspective-transformed coordinate system, influenced by the axes of the machine tool. This established direction is represented as angle θ, relative to the positive x-axis of the machine tool. Upon transforming the direction of the melt-pool deposition into the image coordinate system, the melt-pool width information can be directly extracted based on pixel relationships, as depicted in Figure 2b. Directional correction entails utilizing an affine transformation matrix, expressed as follows:(16)$$T_{p}=\left[\begin{array}{ccc} 1 & 0 & 0\\ 0 & 1 & 0\\ t_{x} & t_{y} & 1 \end{array}\right],$$
(17)$$T_{r}=\left[\begin{array}{ccc} \cos\theta & \sin\theta & 0\\ -\sin\theta & \cos\theta & 0\\ 0 & 0 & 1 \end{array}\right],$$
where $T_{p}$ is the translation matrix and $t_{x}$ and $t_{y}$ are the translation distances along the x and y axes, respectively. $T_{r}$ is the rotation matrix, and θ is the rotation angle, positive in the counterclockwise direction.

## 3. Melt-Pool Width Measurement Experiment

### 3.1. Experimental Setup

As shown in Figure 6a, the experimental setup for DLD featured a five-axis hybrid additive and subtractive machining center.

Powder material was delivered to the nozzle via argon gas through four tubes and deposited using an LDM2000-60 coaxial laser (Deloitte, Ningbo, China) with a wavelength of 980 nm and a power output of 2 kW. The five-axis system allowed for powder deposition in multiple directions due to its complex motion capabilities. Figure 6b illustrates the binocular vision system utilized for measuring melt-pool dimensions, with camera 1 positioned as the frontal camera and camera 2 as the side camera. Both cameras are MV-GE502C-T color CMOS industrial cameras with a maximum resolution of 2592 × 1944 pixels and a maximum pixel count of 5 million pixels. For simplified adjustment, HC3516-F02 manual zoom lenses were utilized, each equipped with a neutral-density filter offering 5% transmittance and fitted in front of the lens.

### 3.2. Experimental Materials

The experimental substrate material was 316 L stainless steel, measuring 4 mm in thickness and 150 mm in diameter. Prior to DLD, the substrate underwent sandblasting and was cleaned with alcohol and acetone. The main chemical components of the materials are detailed in Table 1.

The powder material is a K418 nickel-based superalloy featuring a particle diameter ranging from 15 to 53 µm, with powder flowability measured at 16.4 s and 50 g, indicative of good flowability. Renowned for its outstanding high-temperature and oxidation resistances, this material finds extensive application in the aerospace and petrochemical industries. The primary chemical components of the material are outlined in Table 2.

### 3.3. Experimental Details

#### 3.3.1. Melt-Pool Width Measurement Experiment for Different Deposition Directions

To assess the efficacy of the proposed binocular vision measurement method, experiments were devised to detect melt pools in diverse deposition directions. This involved adjusting the advance angle of the melt pool in 15° increments, specifically to 15°, 30°, 45°, 60°, and 75°, as depicted in Figure 7.

The optimal process parameters, informed by the multiobjective optimization results outlined in [27], encompassed laser power (1100 W), scanning speed (8 mm/s), and powder feeding speed (0.25 r/min). These parameters were selected to execute melt-pool measurement experiments across various deposition paths.

#### 3.3.2. Accuracy Verification Method

In the DLD process, the melt pool remains in a high-temperature liquid state, posing a challenge for accurately verifying machine-vision-based online measurement methods. Typically, accuracy verification relies on offline measurements of the cooled deposited layer. However, a notable disparity exists between the width of the high-temperature liquid melt pool and that of the solidified deposited layer post-cooling, making direct comparison unfeasible. To validate the accuracy of the proposed binocular vision online measurement method, the relationship between the melt-pool width and the deposited layer width must be established.

Accordingly, a mapping model correlating the melt-pool width with the deposited layer width was developed through various DLD experiments conducted under different process parameters. Based on initial experimental findings, eight sets of typical process parameters were selected (detailed in Table 3: laser power (*P*), scanning velocity (*V*_s_), and powder mass flow rate (*V*_f_)) with deposits oriented in the 0° direction, as depicted in Figure 7. The melt-pool width was detected online using a side-mounted monocular camera.

Measurements of the deposited layer were performed utilizing an optical gaging product (OGP) optical measuring device, boasting an accuracy of 1.2 + L/250 microns. Following laser deposition, the substrate was situated on the displacement stage. The intersections where the deposited layer intersected the substrate were scanned using a low-angle ring light and linearly analyzed to ascertain the width of the deposited layer, as depicted in Figure 8.

The online and offline measurement results were linearly correlated to establish a mapping model from the melt pool to the deposited layer. The binocular vision multiangle online measurement results were then fed into this mapping model to compute the corresponding deposited layer widths. Subsequently, these widths were juxtaposed with actual offline measurements obtained from the OGP to validate the accuracy of the binocular vision measurement method.

## 4. Experimental Results

### 4.1. Melt-Pool Width Measurement Results for Different Deposition Directions

The process of online measurement of melt pools in various directions using binocular vision is depicted in Figure 9. As elucidated in Section 2.1, the contours of the melt pool before transformation, illustrated in Figure 9a,c, signify that the near edge represents the true contour, whereas the far edge signifies the false contour. Following transformation, the true contours are depicted at the lower edge in Figure 9b and the upper edge in Figure 9d. To accurately extract width information, correcting the deposition direction and subsequently merging the two true contours are essential, as demonstrated in Figure 9e.

For each deposition angle, three high-quality frames were captured at one-second intervals for measurement, with the average of these frames employed as the final result. The final results of the melt-pool width measurements are summarized in Table 4.

### 4.2. Establishment of Mapping Model from Melt-Pool Width to Deposited Layer

In the online monocular vision measurement process, as the deposition direction aligns with the camera layout, extracting melt-pool width information is easily achieved using the minimum bounding rectangle method, as depicted in Figure 10. Similarly, three high-quality frames were captured during the deposition process at one-second intervals for averaging.

The offline measurements of the deposited layer are shown in Figure 11b–d. These panels display the widths of the deposited layers measured utilizing the OGP optical measuring device under various process parameters, as detailed in Table 3 (numbers 2, 3, and 4).

Table 5 presents the conclusive results of melt-pool widths (W_m_) and deposited layer widths (W_d_) for eight parameter groups. Evidently, the melt-pool width consistently appears smaller than the deposited layer width, yet both display analogous trends, aligning with the findings of reference [12]. Utilizing linear fitting via the least squares method, a mapping model from melt-pool width to deposited layer width is formulated, as depicted in Figure 12.

### 4.3. Accuracy Verification Results

Figure 13 illustrates the offline measurement results of the deposited layer at various deposition angles, where each deposited layer spans a length of 40 mm.

The online measurement results (W_m_) ranging from 15° to 75°, as listed in Table 4, were utilized as input into the mapping model to predict the deposited layer widths (W_p_). These predicted widths were then compared with the offline-measured widths (W_d_) to compute deviations and validate accuracy, as outlined in Table 6. The findings reveal that the measurement method employing binocular cameras exhibits an absolute error of 0.034 mm at a 45° deposition angle, closely aligning with the offline measurement results from the OGP, demonstrating a linear error of 1.19%. Across the measurement range of 30°–60°, the average error remained below 3%. For melt pools advancing at 15° and 75°, the maximum error percentage reached 12.81%. The final accuracy verification results are presented in Table 6.

Analysis of the data distribution revealed that the binocular vision measurement method described in this paper demonstrated high accuracy within the deposition angle range of 30–60°. However, the measurement error progressively increased at deposition angles of 15° and 75°, as shown in Figure 14.

This trend corresponds to the hardware composition and algorithm design principles of the measurement system. In terms of hardware, the binocular cameras are positioned at 90° to each other, enabling each camera to capture part of the true edges of the melt pool from both sides. Consequently, within the central range of camera angles centered at 45°, cameras on both sides are equally assigned the true edge of the melt pool. However, for melt pools moving at 15°, the edges captured by the side-mounted camera were significantly shortened; similarly, at 75°, the edges captured by the front-mounted camera were significantly shortened, thereby reducing the reliability of the edge information obtained and incrementally increasing the measurement error. At the algorithmic level, the perspective transformation applied to the original image induces a stretching effect, which amplifies the measurement error of the small edges at extreme angles.

Figure 15 illustrates the principle behind this phenomenon caused by the camera distribution in the hardware setup. As the angle increases from 45° to 75°, the proportion of the true lower edge captured by the front camera continuously decreases, particularly at the widest edge of the melt pool, which reflects the width information. Increasing the intercamera angles to improve the camera layout can reduce the occurrence of critical edge information entering the blind spots to a certain extent. Furthermore, within the constraints of the available space in the equipment, increasing the number of cameras and selecting the optimal camera for detection at different angles can significantly enhance the detection range and accuracy.

## 5. Conclusions

This study proposed an off-axis binocular vision method for detecting melt-pool width using binocular CMOS cameras to capture images from both sides of the melt pool. The method integrated image denoising, edge recognition, and dimension extraction algorithms. Following DLD experiments, a mapping model was established, combining online melt-pool measurements with offline-deposited layer measurements. This model was employed to verify the accuracy of binocular vision measurements at various deposition angles. The results demonstrated that the method maintained a measurement error within 3% for deposition angles ranging from 30° to 60°, thus extending the measurement range of melt-pool widths. Consequently, it is suitable for integration into closed-loop control systems in laser direct deposition processes, thereby laying the groundwork for enhancing the quality of deposited layers.

In the process of DLD, real-time, accurate measurement of any direction of the melt-pool size is critical to ensure the quality of the cladding layer [28]. This study focuses on the off-axis vision inspection method for detecting melt-pool width within a certain angle range in single-pass single-layer deposition. In future research, the rapid measurement of multichannel and multilayer melt pools with arbitrary deposition angles needs to be studied more deeply.

## Figures and Tables

**Figure 1 materials-17-03337-f001:**
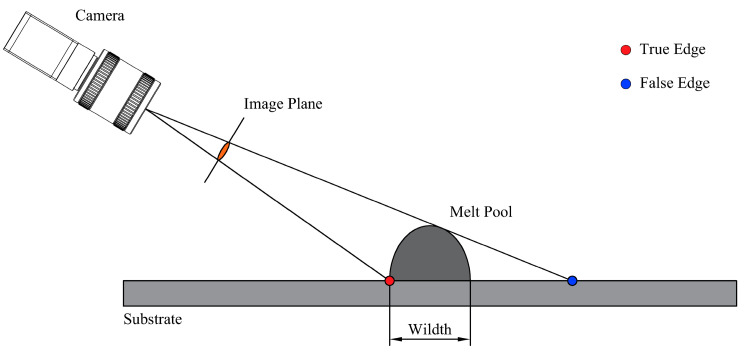
Schematic of melt-pool imaging.

**Figure 2 materials-17-03337-f002:**
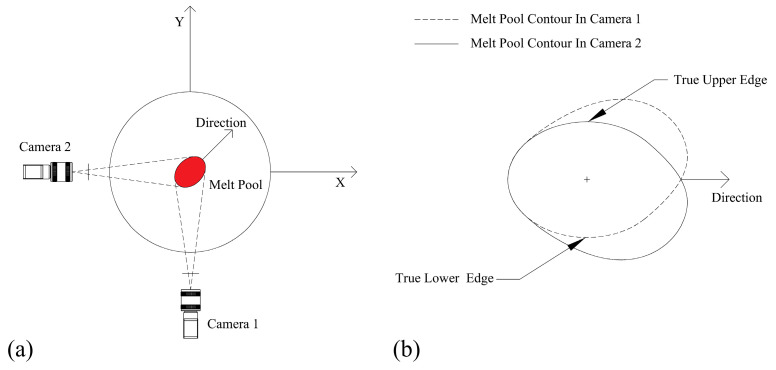
Principle of binocular vision melt-pool measurement. (**a**) Binocular cameras detect the edges of the melt pool on both sides; (**b**) combined extraction of the melt-pool width.

**Figure 3 materials-17-03337-f003:**
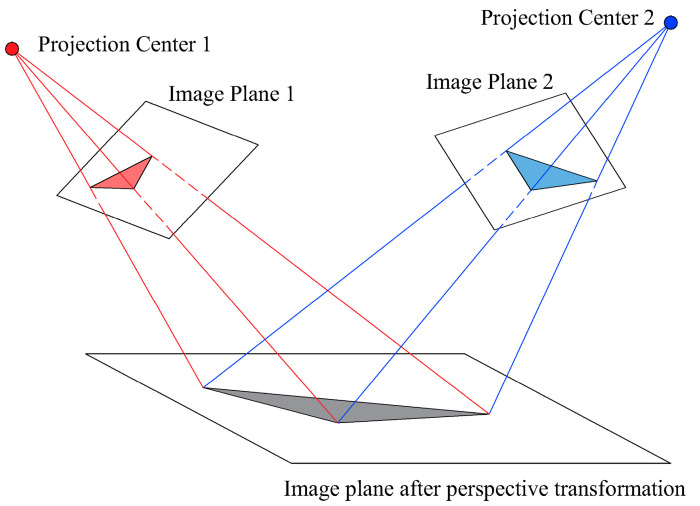
Principle of perspective transformation.

**Figure 4 materials-17-03337-f004:**
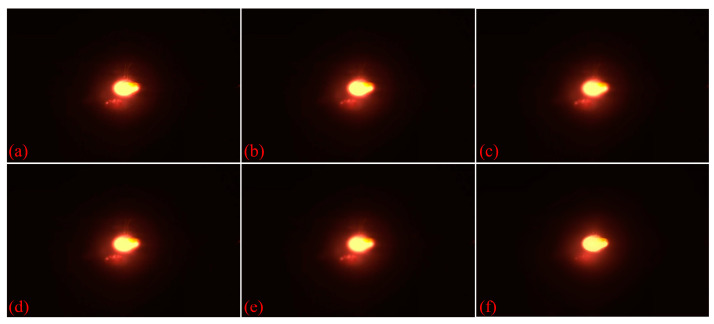
Comparison of five common filtering methods: (**a**) Original image, (**b**) median filtering, (**c**) box filtering, (**d**) Gaussian filtering, (**e**) mean filtering, and (**f**) bilateral filtering.

**Figure 5 materials-17-03337-f005:**
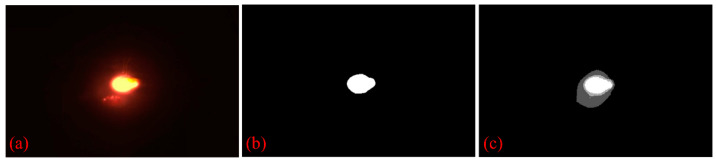
Threshold segmentation comparison: (**a**) Original image, (**b**) Otsu threshold segmentation, and (**c**) multilevel fuzzy entropy threshold segmentation.

**Figure 6 materials-17-03337-f006:**
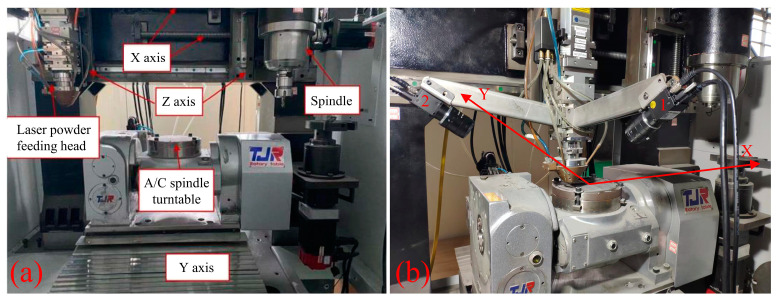
Experimental setup. (**a**) Five-axis additive and subtractive composite machining center; (**b**) binocular vision melt-pool width measurement system.

**Figure 7 materials-17-03337-f007:**
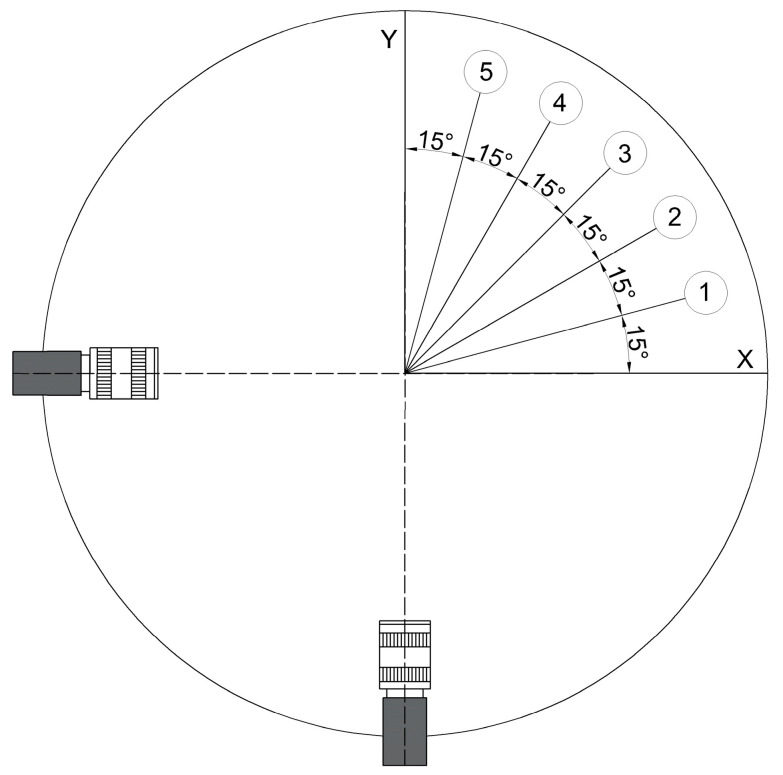
Schematic of the experiment for different deposition directions.

**Figure 8 materials-17-03337-f008:**
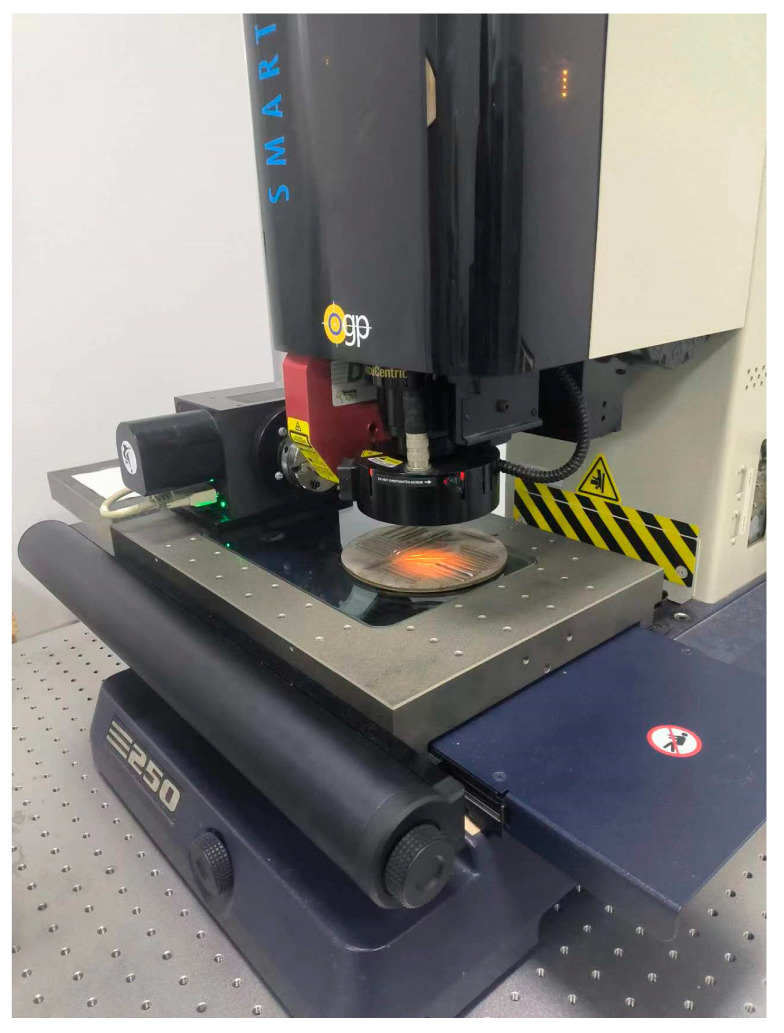
Photograph showing the offline measurement process of deposited layer width.

**Figure 9 materials-17-03337-f009:**
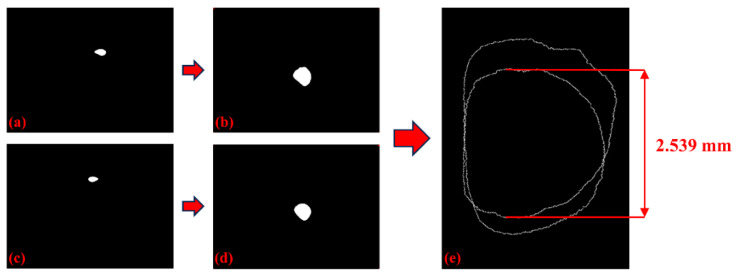
Binocular vision melt-pool measurement process: (**a**) Front camera before transformation, (**b**) front camera after transformation, (**c**) side camera before transformation, (**d**) side camera after transformation, and (**e**) width extraction using affine transformation angle correction.

**Figure 10 materials-17-03337-f010:**
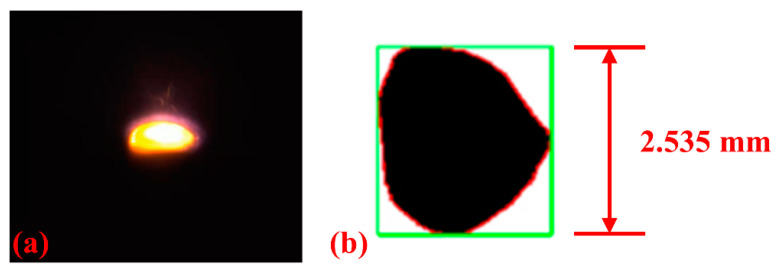
Monocular vision melt-pool measurement: (**a**) Melt-pool image and (**b**) melt-pool width calculation.

**Figure 11 materials-17-03337-f011:**
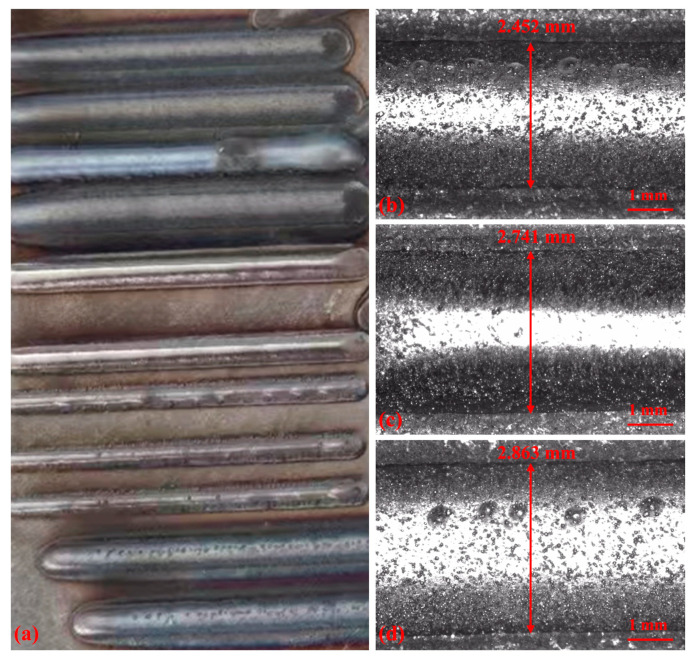
Deposited layer and OGP measurement images under different process parameters: (**a**) Deposited layer, (**b**) number 2, (**c**) number 3, and (**d**) number 4.

**Figure 12 materials-17-03337-f012:**
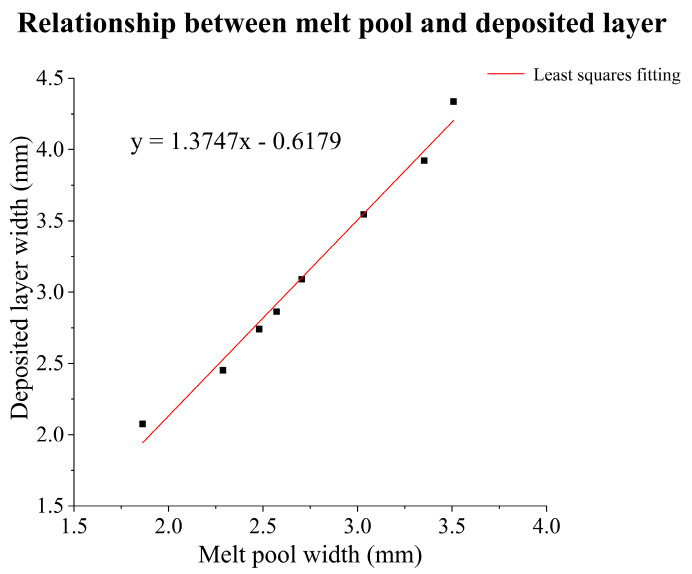
Mapping model from melt-pool width to deposited layer width.

**Figure 13 materials-17-03337-f013:**
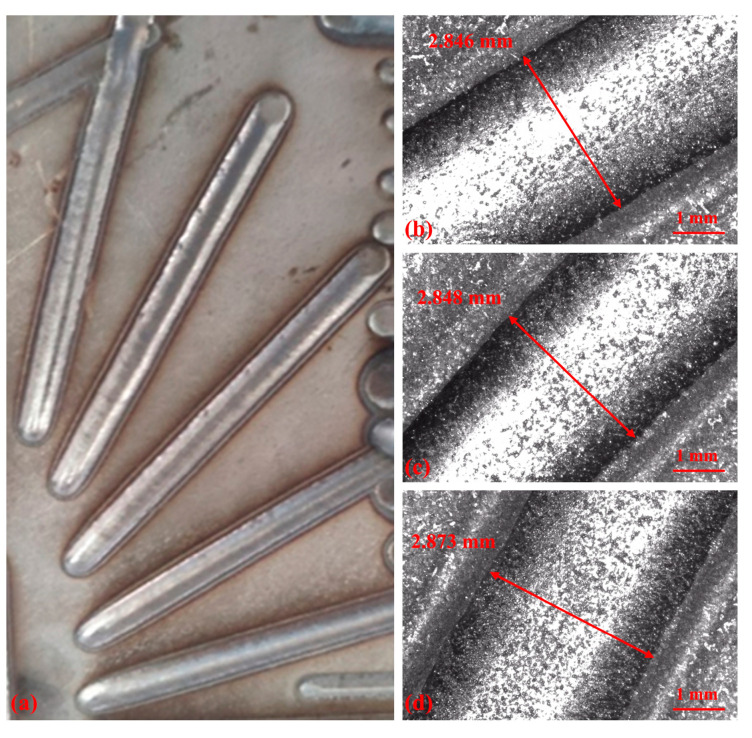
Deposited layer and OGP measurement images under different deposition angles: (**a**) Deposited layer image, (**b**) 30°, (**c**) 45°, and (**d**) 60°.

**Figure 14 materials-17-03337-f014:**
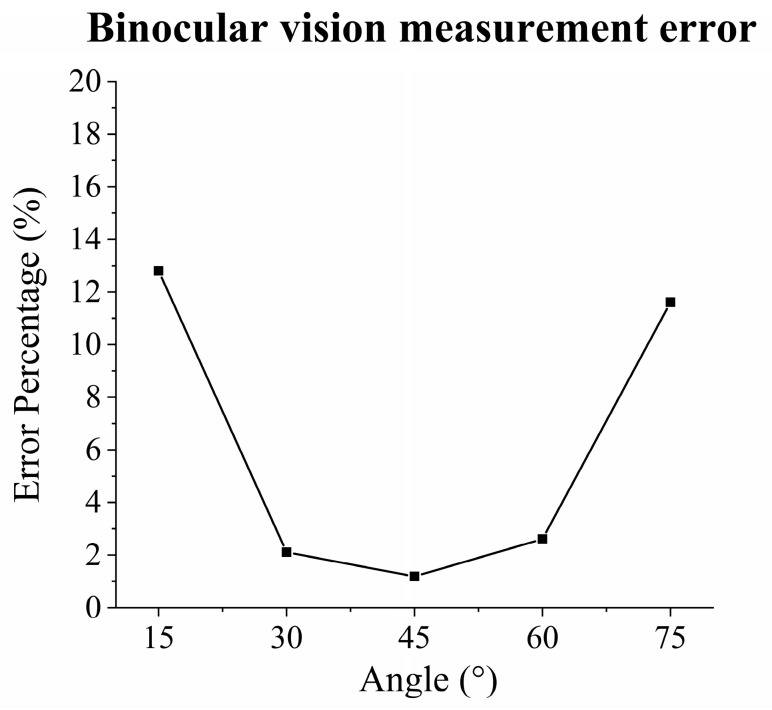
Binocular vision measurement error.

**Figure 15 materials-17-03337-f015:**
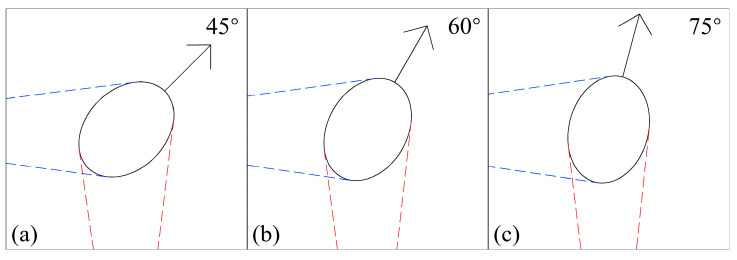
Effect of the change in melt-pool deposition angle on edge acquisition: (**a**) 45°, (**b**) 60°, and (**c**) 75°.

**Table 1 materials-17-03337-t001:** Chemical composition (mass fraction) of K418 nickel-based alloy powder (%).

Ni	Cr	Mo	Ti	Al	C	Si
Bal	12.61	4.06	0.95	6.31	0.10	0.12
Mn	P	S	Fe	Nb	O	N
0.01	0.01	0.002	0.05	2.10	0.01	0.005

**Table 2 materials-17-03337-t002:** Chemical composition (mass fraction) of 316 L stainless steel (%).

Cr	Ni	Cu	Mn	Si	Mo	Nb	C	Fe
15–17.5	3–5	3–5	≤1	≤1	≤0.5	0.35	≤0.07	bal

**Table 3 materials-17-03337-t003:** Process parameter group.

Number	*P* (W)	*V*_s_ (mm/s)	*V*_f_ (r/min)
1	900	11	0.45
2	1100	9	0.55
3	1100	7	0.25
4	1100	8	0.25
5	900	3	0.15
6	1100	3	0.15
7	1300	3	0.15
8	1500	3	0.15

**Table 4 materials-17-03337-t004:** Results of melt-pool width measurement by binocular cameras.

Number	Angle (°)	Width (mm)
1	15	2.761
2	30	2.476
3	45	2.546
4	60	2.485
5	75	2.834

**Table 5 materials-17-03337-t005:** Monocular camera melt-pool width measurement results.

Number	W_m_ (mm)	W_d_ (mm)
1	1.863	2.075
2	2.288	2.452
3	2.480	2.741
4	2.572	2.863
5	2.705	3.090
6	3.033	3.546
7	3.353	3.923
8	3.508	4.338

**Table 6 materials-17-03337-t006:** Final accuracy verification results.

Number	Angle (°)	W_m_ (mm)	W_p_ (mm)	W_d_ (mm)	Error (mm)	Error (%)
1	15	2.761	3.178	2.817	0.361	12.81%
2	30	2.476	2.786	2.846	0.06	2.11%
3	45	2.546	2.882	2.848	0.034	1.19%
4	60	2.485	2.798	2.873	0.075	2.61%
5	75	2.834	3.278	2.937	0.341	11.61%

## Data Availability

The original contributions presented in the study are included in the article, further inquiries can be directed to the corresponding author.

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
