# Peer review of "Online Measurement of Melt-Pool Width in Direct Laser Deposition Process Based on Binocular Vision and Perspective Transformation"

_materials, 2024, doi:10.3390/ma17133337_

Round 1

Reviewer 1 Report

Comments and Suggestions for Authors

The inclusion of relevant research and analyses in future work can contribute to further improvement of the proposed method and increase its acceptance in practical industrial applications. This will improve the versatility and accuracy of the measuring system in different conditions.
Improvement proposals: (i) Figure 12 and 14 should have an enlarged font describing the axis for a better presentation of the results, (ii) The introduction of additional cameras or sensors that can complement the current layout can also help in obtaining a more complete image of the lake, regardless of the angle.

Author Response

Dear Reviewers,  

Thank you very much for your time spent reviewing the manuscript and making encouraging comments on its merits. We would like to take this opportunity to thank you for your time and this great opportunity for us to improve the manuscript. We hope you will find this revised version satisfactory.

We have responded to your comments point by point in the Word document uploaded in the attachment.

We would like to take this opportunity to thank you for your time and this great opportunity for us to improve the manuscript. We hope you will find this revised version satisfactory.  

Sincerely,

The Authors

Reviewer 2 Report

Comments and Suggestions for Authors

The paper presents a method based on machine vision and image segmentation, for the online measurement of melt pool width in direct Laser deposition process.

The measurement procedure is mainly based on:

-         Binocular vision

-         Perspective transformation

-         Image segmentation based on fuzzy entropy threshold.

The references are appropriate and the method proposed by the authors introduces significant improvements respect the other methods available in open literature, based on monocular vision and/or affected by image noise.

The figures and the numerical model formulas are clear.

The conclusions well summarize the results achieved by the authors.

However, minor revisions are required as reported in the PDF file uploaded.

Author Response

(The authors gave the same response as above.)

Reviewer 3 Report

Comments and Suggestions for Authors

In this work, Lu et al. propose a new technique for real-time monitoring of the melt-pool width in direct laser deposition method.

The novelty of the work is widely explained by the authors, who exhaustively describe the state-of-the-art and the progress introduced beyond it. 

The technique is accurately described from a theoretical point of view, and results are then experimentally validated by a real deposition session.

I just have some suggestions for the authors:

  1. In my opinion, the term “online measurement” used in the title should be replaced with “real-time monitoring”, which is more correct and appealing to the reader.
  2. Lines 9-10. What do the authors mean with “stability” and “portability”? Stability of the deposited layer? Portability of the deposition system?
  3. Line 12. I would replace “machine vision methods” with “camera-assisted inspection methods”.
  4. Eq.(1). Write that “w” is the third coordinate, and that w=1 is used without loss of generality.
  5. Eq.(5). Define “L”.
  6. Eq.(9). Define “P” and “p”.
  7. Line 192. Otsu threshold segmentation is mentioned. I suggest to describe it in a few words, or to provide a suitable reference at least.
  8. The introduced technique proved to be very effective when the angle is in the 30-60° range, but accuracy rapidly drops outside this range. Authors correctly explain why, but I suggest to add a few lines in the concluding remarks describing possible practical solutions to improve accuracy in a wider range of deposition angles.

Author Response

(The authors gave the same response as above.)

Round 2

Reviewer 3 Report

Comments and Suggestions for Authors

The revised version is ok.

Authors have welcomed my suggestions and answered satisfactorily to my comments.